# A Case of Pulmonary Fibrosis and COVID-19-Related Pneumonia in a Pembrolizumab-Treated Patient

**DOI:** 10.3390/idr17030053

**Published:** 2025-05-12

**Authors:** Alberto Zolezzi, Gina Gualano, Annelisa Mastrobattista, Pietro Vittozzi, Virginia Di Bari, Carlotta Cerva, Silvia Mosti, Antonio Lugini, Fabrizio Albarello, Federica Di Stefano, Maria Beatrice Valli, Fabrizio Palmieri

**Affiliations:** 1Respiratory Infectious Diseases Unit, National Institute for Infectious Diseases “Lazzaro Spallanzani” IRCCS, 00149 Rome, Italy; alberto.zolezzi@inmi.it (A.Z.); a.mastrobattista@inmi.it (A.M.); pietro.vittozzi@inmi.it (P.V.); virginia.dibari@inmi.it (V.D.B.); carlotta.cerva@inmi.it (C.C.); silvia.mosti@inmi.it (S.M.); fabrizio.palmieri@inmi.it (F.P.); 2Medical Oncology, Azienda Ospedaliera San Giovanni Addolorata, 00184 Rome, Italy; a.lugini@hsangiovanni.roma.it; 3Diagnostic Imaging Unit for Infectious Diseases, National Institute for Infectious Diseases “Lazzaro Spallanzani” IRCCS, 00149 Rome, Italy; fabrizio.albarello@inmi.it (F.A.); federica.distefano@inmi.it (F.D.S.); 4Laboratory of Virology and Biosafety Laboratories, National Institute for Infectious Diseases “Lazzaro Spallanzani” IRCCS, 00149 Rome, Italy; mariabeatrice.valli@inmi.it

**Keywords:** pembrolizumab, severe acute respiratory syndrome coronavirus 2, fibrotic-like pattern

## Abstract

Pembrolizumab is used as a first-line treatment of non-small cell lung cancer. Pneumonitis and interstitial lung disease are among the most common immune-related adverse events. The impact of severe acute respiratory syndrome coronavirus 2 (SARS-CoV-2) infections on patients with cancer treated with chemotherapy or immune checkpoint inhibitors (ICIs) is not fully known. Blocking immune checkpoints may conversely augment dysfunctional T-cell responses in severe patients and, in turn, mediate immunopathology. Here, we present a case of SARS-CoV-2 infection complicated by acute respiratory distress syndrome (ARDS) and a fibrotic-like pattern in a patient treated with pembrolizumab for lung cancer. The patient showed a dramatic clinical and radiological response after steroid therapy. Further research is needed to better understand the long-term implications of pembrolizumab therapy in patients recovering from coronavirus disease 2019 (COVID-19) and to develop evidence-based guidelines for managing these complex cases. Patients undergoing oncologic immunotherapy might benefit from early high-dose steroid treatment in cases of viral infections, such as SARS-CoV-2.

## 1. Introduction

The introduction of immune checkpoint inhibitors (ICIs) therapy led to a significant improvement in the survival of various tumors [1]. Pembrolizumab has been found to be superior to other chemotherapeutic agents as first-line treatment in metastatic non-small cell lung cancer (NSCLC) [2].

Immune-related adverse events (irAEs) have been reported at a prevalence of about 20–30%, showing various clinical manifestations, such as pneumonitis, skin reactions, endocrinologic diseases, colitis, hepatitis, and infusion reactions.

Of these irAEs, pneumonitis, including interstitial lung disease (ILD), is clinically significant and can be life-threatening, thus requiring immediate clinical attention. The incidence of ICIs-related ILD is approximately 4% in patients treated with programmed death-1 (PD-1) inhibitors (such as nivolumab and pembrolizumab) and 2% in those treated with programmed death ligand 1 (PD-L1) inhibitors (such as atezolizumab and durvalumab) [3]. Checkpoint inhibitor-related pneumonitis (CIP) in patients with NSCLC is reported to have an incidence of up to 19% for all grades, and the routine treatment of CIP is based on systemic corticosteroids [4,5].

The impact of SARS-CoV-2 infections on cancer patients receiving chemotherapy or ICIs remains unclear [6]. Patients with cancer who become infected with COVID-19 can experience worse outcomes from their infection because of their immunosuppressive status, which is a result of anti-cancer treatments, including chemotherapy and radiotherapy, as well as from tumor itself [7]. A monocentric retrospective study published in 2022 suggests that COVID-19 may pose a risk of severe irAEs in cancer patients receiving ICIe [6]. Another study reported that ICIs use was not associated with increased risk for COVID-19 death [8]. In a study of 110 patients with laboratory-confirmed SARS-CoV-2 who were undergoing ICIs treatment, 23 patients developed respiratory failure. The authors concluded that COVID-19-related mortality in the ICIs-treated population did not appear to be higher than previously published mortality rates for patients with cancer [9].

Here, we present a case of COVID-19-related moderate acute respiratory distress syndrome (ARDS) [6] complicated by a fibrotic-like pulmonary pattern in a patient treated with pembrolizumab for lung cancer.

## 2. Detailed Case Presentation

A 66-year-old Italian man presented to our institution for COVID-19, pneumonia, and respiratory failure in November 2023.

In September 2020, computed tomography (CT) of the lung showed a tumor mass with a diameter of 63 mm in the left superior lobe (as shown in Figure 1). He was diagnosed with pulmonary adenocarcinoma (TTF1+, PD-L1 > 90%, ALK negative) [10] and was treated after diagnosis with immunotherapy (pembrolizumab 200 mg/3 weeks). Emphysema was also visible. In July 2023, the lung CT scan showed a decrease in tumor mass dimension in the left 24 × 16 mm (as shown in Figure 2A) without signs of interstitial involvement.

The patient had been vaccinated twice against COVID-19, had a history of smoking (30 pack-years) and was receiving ongoing treatment for systemic hypertension and dyslipidemia. On 7 November 2023, three days after the pembrolizumab administration, the patient developed fever (the temperature was 38.1 degrees), a syncopal episode, and head trauma and was admitted to the emergency department.

A lung CT scan performed on the same day was negative for traumatic injury but showed bilateral ILD with “ground glass” and “crazy paving” areas, particularly in the lower lobes. In the apical–dorsal segment of the right superior lobe, a 15 mm consolidation was observed, corresponding to the previously identified neoplasm (as shown in Figure 2B). A naso-pharingeal swab (NPS) tested positive for SARS-CoV-2 infection. SARS-CoV2 RNA was identified in all respiratory samples using a qualitative real-time reverse-transcriptase polymerase chain reaction (RT-PCR) assay. Qualitative real-time RT-PCR tests for influenza A and B viruses, as well as respiratory syncytial virus, were negative. Blood tests for HIV 1 and 2 were also negative. Urinary antigens for Legionella and Pneumococcus returned negative results. Due to worsening clinical conditions, only microbiological data from sputum were available. Sputum examination was negative for common germs and mycobacteria. Blood chemistry tests showed an increased reactive C protein (RCP) of 21.24 mg/dL and a relative neutrophilia at blood count of 81.7%, with a normal white cell count (5.780/uL). The patient was transferred to our department and received steroid treatment (40 mg of methylprednisolone intravenous daily) oxygen support, 2 g of ceftriaxone intravenous daily, and remdesivir, according to local protocol at the time [11].

At hospital admission, the pO_2_/FiO_2_ (P/F) ratio [12] was 236. On day 7, respiratory failure worsened (P/F ratio 144) and non-invasive ventilation (NIV) was initiated, using a full-face mask and bi-level positive airway pressure with Trilogy Evo (Respironics ^®^, Inc., 1010 Murry Ridge Ln, Murrysville, PA 15668, USA) ventilator.

The lung CT scan performed on day 9 (as shown in Figure 2B) showed a fibrotic-like pulmonary pattern with consolidation in the lower lobes and a worsening of the crazy paving areas; the patient was switched to a high-dose steroid treatment of 1 mg per kg/die of methylprednisolone, according to ARDS protocol [13,14], and 13.5 mg iv/die of piperacilline/tazobactam, administered for 14 days. On day 11, the SARS-CoV-2 NPS tested negative. Pembrolizumab was discontinued.

Soon after, the high-dose steroid treatment clinical status improved. On 15 December 2023, the NIV treatment was suspended (P/F 258 in FiO_2_ 31%), and after 61 days, the patient was discharged under treatment of long-term oxygen therapy and systemic steroid therapy (prednisone 25 mg), with a reduction every 10 days.

After discharge, the patient was monitored by oncologists during follow-up. He discontinued pembrolizumab after PF events (after 37 months of therapy) and achieved complete remission of lung cancer.

CT performed after discharge on February 2024 showed a dramatic improvement of ILD; the “crazy paving” areas were reduced and the inferior lobe consolidations, where traction bronchiectasis arose, had disappeared (as shown in Figure 2C).

The patient was alive and in complete remission 18 months after stopping therapy.

## 3. Discussion

Here, we present a rare case of SARS-CoV-2 infection complicated by pulmonary fibrosis (PF) [15] in a patient treated with pembrolizumab for lung cancer. In the literature, we found few papers (four) on detailed cases of COVID-19 pneumonia in conjunction with ICIs pneumonitis [6,16,17,18].

The COVID-19 pandemic highlighted significant challenges for the management of cancer patients, particularly those undergoing immunotherapy. Pembrolizumab, an ICI-targeting PD-1, has been widely used in the treatment of various cancers [1,19]. However, concerns have arisen regarding its use in patients who have recovered from COVID-19, due to the potential for exacerbating (irAEs) or influencing COVID-19 sequelae [20]. Pembrolizumab is a monoclonal antibody that blocks the PD-1 receptor, which is critical for regulating immune tolerance and enhancing the immune response against cancer cells. It has become a cornerstone therapy for multiple malignancies, including NSCLC, melanoma, and head and neck squamous cell carcinoma [2]. However, this heightened immune activity can lead to irAEs, which can affect multiple organ systems, including the lungs, liver, and endocrine glands [3,5]. The COVID-19 pandemic added complexities to managing patients on pembrolizumab, particularly in terms of the long-term effects of COVID-19 and the potential risks of exacerbating these effects with ongoing immunotherapy [20]. There is concern that pembrolizumab may worsen post-COVID-19 inflammatory conditions, such as PF, myocarditis, or other autoimmune phenomena [20]. In a recent study, Tanvetyanon and colleagues [21] found that the survival of patients who underwent frontline pembrolizumab-based treatment was better in the pandemic cohort than in the pre-pandemic cohort, when the PD-L1 expression level was <50%; survival did not worsen if PD-L1 level was ≥50%. However, the study is limited by its non-randomized design, and the data were obtained during routine clinical care, not a controlled research environment, and thus subjected to delays or other issues, such as missing data. The case described in our paper may support these researchers’ hypothesis, as the clinical outcome of the patient was favorable. However, further prospective studies are necessary to confirm the hypothesis.

Clinicians should be fully aware of the possibility of ICIs-induced pulmonary toxicity when using PD-1/PD-L1 inhibitors. CIP is one of the most common fatal adverse events of PD-1/PD-L1 inhibitors [4]. CIP can be controlled by routine corticosteroid treatment in approximately 70% to 80% of patients [4].

CT findings of ICIs-related pneumonitis vary, often showing patchy GGOs and consolidating, symmetrical distributions, predominantly in the lower lobes and peripheral areas. Patterns can include non-specific interstitial pneumonia, organizing pneumonia, diffuse alveolar damage, hypersensitivity pneumonitis, or simple pulmonary eosinophilia, appearing in conjunction with COVID-19 pneumonia.

The criteria for diagnosing drug-related pneumonitis include (a) newly detected parenchymal opacities on CT or chest radiographs, usually bilateral and non-segmental; (b) temporal association with a systemic therapeutic agent; and (c) the exclusion of other causes [22].

Approximately 40% of COVID-19 pneumonia patients develop ARDS [23], and it is well known that patients hospitalized for COVID-19 can often develop persistent radiographic abnormalities (1–4), consisting of non-fibrotic, usually ground glass opacities or fibrotic-like traction bronchiectasis, reticulations, and rarely, honeycombing [24]. Chest CT findings in COVID-19 patients commonly include peripheral, bilateral, or multifocal GGOs with or without consolidation near pleural surfaces, fissures, or visible interlobular lines, creating a “crazy paving” pattern. Later stages may show reverse halo and other organizing pneumonia signs [25].

We observed that the patient had developed fibrotic-like patterns with consolidation in lower lobes and the worsening of “crazy paving” areas. COVID-19, pneumonia, and pembrolizumab-induced pneumonitis can share immune-related effects, raising the possibility of combined immune-related toxicity. A monocentric study published in 2022 suggests that COVID-19 may pose a risk of severe irAEs in cancer patients receiving ICI. Close monitoring and possibly delaying ICIs administration could be considered when cancer patients are infected with COVID-19 [6]. COVID-19 infection increases pro-inflammatory cytokines, leading to cytokine release syndrome (CRS), characterized by increased levels of interferon (IFN)-γ, interleukin-6 (IL-6), and other cytokines, which results in immune hyperactivation. CRS has been reported in patients receiving ICIs and in those with COVID-19 infections [16].

The clinical similarity between pneumonitis caused by ICIs and pneumonia associated with COVID-19 has posed considerable challenges for cancer patients and oncologists [16,18]. COVID-19 pneumonia and pembrolizumab-induced pneumonitis present with non-specific symptoms such as cough, dyspnea, fever, and fatigue and, radiologically, both can exhibit ground-glass opacities and bilateral infiltrates in chest CT scans [18]. Given the intersecting features, a differential diagnosis can be challenging, particularly in areas with high COVID-19 prevalence.

A thorough patient history, including recent exposure to COVID-19 and the timing of symptom onset relative to pembrolizumab administration, is crucial. Laboratory findings, such as lymphopenia, elevated inflammatory markers (e.g., RCP, ferritin), and COVID-19 polymerase chain reaction (PCR) testing, can assist in the diagnosis. However, these tests are not definitive, and false negatives in COVID-19 PCR testing can complicate the clinical picture.

Given the potential risks of worsening, adopting a multidisciplinary approach to manage cancer patients who receive pembrolizumab after recovering from COVID-19 is crucial. Regular monitoring for signs of irAEs, particularly in organs previously affected by COVID-19, is essential. In some cases, delaying pembrolizumab therapy or adjusting the dosage might be necessary, especially in patients with significant post-COVID-19 pulmonary or cardiac sequelae [18]. The intersection of pembrolizumab treatment and COVID-19 sequelae presents a new frontier in cancer care that requires careful consideration. While pembrolizumab remains a vital tool in the oncologist’s arsenal, its use in the post-COVID-19 context demands heightened vigilance and a tailored approach to each patient’s unique clinical scenario. Further research is needed to better understand the long-term implications of pembrolizumab therapy in patients recovering from COVID-19 and to develop evidence-based guidelines for managing these complex cases [18]. A cross-sectional study in China suggests that lung cancer patients receiving ICIs and experiencing irAEs may have a higher risk of developing COVID-19 pneumonia due to the Omicron variant. Therefore, the close monitoring of these patients during COVID-19 is necessary to mitigate this risk [20]. Patients receiving ICIs should be vaccinated, and strict mitigation strategies should be maintained in cancer centers to minimize nosocomial transmission [18]. Prompt NPS execution to combat fever or inflammatory symptoms and serum inflammation indices can aid differential diagnosis. Guidelines are needed for managing cases of COVID-19 pneumonia and pembrolizumab-induced pneumonitis, including for the use of immediate systemic steroid therapy, which benefited our patient.

## 4. Conclusions

PD-1 inhibitor immunotherapy and other ICIs revolutionized lung and other cancer treatments. ILD is among the adverse effects. We believe that COVID-19 can trigger or quicken PF initiation but our patient might have had a predisposition to PF due to ongoing pembrolizumab therapy. Also, the patient’s cancer might have caused a predisposition toward more severe COVID-19 manifestations. Further studies could outline the role of other antifibrosant drugs in PF, but at the moment, an immediate steroid treatment seems effective—as was demonstrated in this case. Our case will contribute to the general discussion and to further reviews and analysis. Further research is needed to better understand the long-term implications of pembrolizumab therapy in patients recovering from COVID-19 and to develop evidence-based guidelines for managing these complex cases [18] and beginning valid treatment.

## Figures and Tables

**Figure 1 idr-17-00053-f001:**
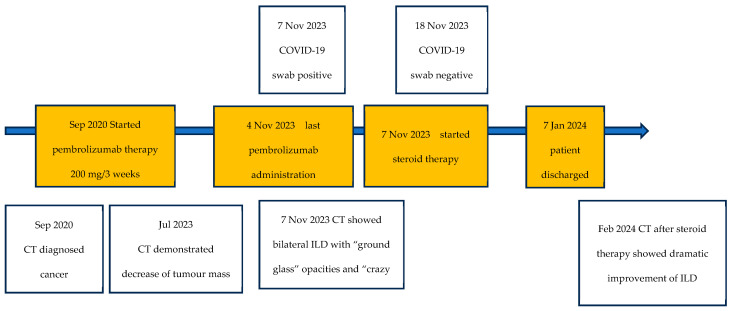
Timeline shows the chronology of imaging scans, COVID-19 swab results, and therapies from the diagnosis of cancer and COVID-19.

**Figure 2 idr-17-00053-f002:**
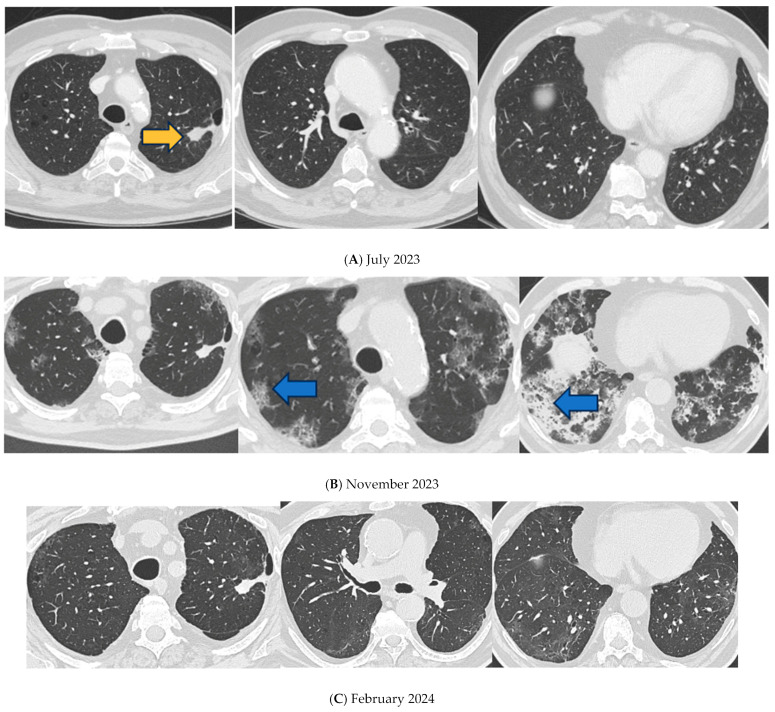
Lung CT scan. (**A**) July 2023. Tumor mass, 24 × 16 mm, at left superior lobe without signs of interstitial involvement (orange arrowhead). (**B**) 17 November 2023. Tumor mass, 15 mm, at left superior lobe with signs of interstitial involvement with “ground glass” and “crazy paving” areas (blue arrowheads), in particular in lower lobes. (**C**) February 2024. Crazy paving interstitiopathy areas were reduced and the of inferior lobes consolidations, located where traction bronchiectasis had arisen, disappeared.

## Data Availability

All relevant data are within the manuscript. Raw data are accessible, if requested, from the National Institute for Infectious Diseases “L. Spallanzani” Library, at the following e-mail address: biblioteca@inmi.it.

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
