# Peer review of "A Case of Pulmonary Fibrosis and COVID-19-Related Pneumonia in a Pembrolizumab-Treated Patient"

_2036-7449, 2025, doi:10.3390/idr17030053_

Round 1
Reviewer 1 Report
Comments and Suggestions for Authors
This manuscript addresses an important and timely clinical scenario with a patient on immune checkpoint inhibitor pembrolizumab, who developed severe COVID-19 pneumonia with a fibrotic-like pattern. The case is rare and very useful to document in detail. The report’s strengths include a clear timeline of events from cancer diagnosis through COVID-19 infection and recovery, the management steps, and the interplay between COVID-19 and immunotherapy-related lung toxicity. The case is clinically informative and provides a useful approach if an aggressive corticosteroid therapy may lead to good outcomes in a complex situation such as this one.
I suggest including any other relevant medical history such as smoking or comorbidities. Also, mentioning the patient’s vaccination status for COVID-19 could be a useful context. It would be helpful to explicitly state the oncologic response status like partial remission of lung cancer under pembrolizumab treatment to underscore the benefit the patient was deriving from immunotherapy.
Author Response
comment 1:I suggest including any other relevant medical history such as smoking or comorbidities. Also, mentioning the patient’s vaccination status for COVID-19 could be a useful context. It would be helpful to explicitly state the oncologic response status like partial remission of lung cancer under pembrolizumab treatment to underscore the benefit the patient was deriving from immunotherapy.
Answer:
We want to thank the reviewer for the appreciation and advice.
The patient was vaccinated 2 times against COVID-19, had a history of smoking (30 pack-years) and was receiving ongoing treatment for systemic hypertension and dyslipidemia.
After discharge, the patient was followed by oncologists. He did not receive further treatment with pembrolizumab and achieved complete remission of lung cancer.
We added the informations in the text on lines 70 and 99.

Reviewer 2 Report
Comments and Suggestions for Authors
The article titled: "A case of pulmonary fibrosis Covid-19 pneumonia related in a 2 pembrolizumab treated patient" written by Zolezzi A is interesting. However, the text must improve some aspects to be published.
Line 31: Add COVID-19
Line 40: The paragraph needs a reference to support what is being explained.
Line 49: The sentence being discussed requires a reference. The authors should also add a paragraph demonstrating the state of the art of the topic being discussed. It is unknown whether this is the first case described, and they should emphasize the novelty of the article.
Line 50: Reference 6 can be deleted.
Line 54: How was the COVID-19 diagnosis made? What test was used?
Line 60: What dose and how often was pembrolizumab administered?
Line 60: What was the temperature value?
Lines 65-69: The authors should provide more details about the tests performed on the patient. Other variables of interest should be described. Were bacteriological/mycological cultures performed on the patient? Were these studies negative?
For a better understanding of the article, the authors could make a diagram (figure) that addresses the time of study of the patient as well as the most relevant details.
Line 106: Authors should add the following citation : Moffat GT, Hanna L, Hopman W, Fung AS, Gaudreau PO. An assessment of extended pembrolizumab dosing in advanced non-small-cell lung cancer in the COVID-19 pandemic. Immunotherapy. 2023 Aug;15(12):921-932. doi: 10.2217/imt-2022-0257. and discuss this idea in the article.
Line 109: Recently, Tanvetyanon T et al. described that the COVID-19 pandemic was associated with an increase in survival among patients with lower PD-L1 expression who were treated with pembrolizumab monotherapy. The authors should discuss this idea in their article.
Line 149: The article's conclusions are redundant and very long. There are elements that could be incorporated into the discussion of the work. The conclusions should be rewritten.
Line 232: Reference 10 must be updated.
Author Response
Rev 2: results must be improved
The article titled: "A case of pulmonary fibrosis Covid-19 pneumonia related in a 2 pembrolizumab treated patient" written by Zolezzi A is interesting. However, the text must improve some aspects to be published.
Answer:
We want to thank the reviewer for the important and detailed suggestions, we have taken this into account as follows:
Line 31 (key words): Add COVID-19. thanks for suggestion, we added “COVID-19”
Line 40: The paragraph needs a reference to support what is being explained.
thanks for suggestion. We added the reference as suggested: Spain, L.; Diem, S.; Larkin, J. Management of toxicities of immune checkpoint inhibitors. Cancer Treat Rev. 2016;44:51–60. https:// doi.org/10.1016/j.ctrv.2016.02.001.
Line 49: The sentence being discussed requires a reference.
The authors should also add a paragraph demonstrating the state of the art of the topic being discussed.
It is unknown whether this is the first case described, and they should emphasize the novelty of the article.
Thanks for suggestions.
The impact of SARS-CoV-2 infections on cancer patients receiving chemotherapy or immune checkpoint inhibitors (ICIs) remains unclear.
Patients with cancer who get infected with COVID-19 might have a worse outcome from the infection because of their immunosuppressive status determined by anti-cancer treatments, including chemotherapy and radiotherapy and by tumor itself
[7].
A monocentric retrospective study published in 2022 suggests that COVID-19 may pose a risk of severe irAEs in cancer patients receiving ICI. Close monitoring and possibly delaying ICI administration could be considered when cancer patients are infected with COVID-19 [6]
The intersection of pembrolizumab treatment and COVID-19 sequelae presents a new frontier in cancer care that requires careful consideration. While pembrolizumab remains a vital tool in the oncologist's arsenal, its use in the post-COVID-19 context demands heightened vigilance and a tailored approach to each patient's unique clinical scenario.
The effects and progression of viral infection in individuals undergoing immunotherapy are yet to be determined.
To our knowledge, limited data are available on the coexistence of COVID-19 and checkpoint inhibitor-related pneumonitis (CIP) in patients with NSCLC.
Bui and coll reported that ICI use was not associated with increased risk for COVID-19 death [8]
Rogers A and colleagues reported that, among 110 patients with laboratory-confirmed SARS-CoV-2 while on treatment with ICI, 23 patients developed respiratory failure. Authors conclude that COVID-19-related mortality in the ICI-treated population does not appear to be higher than previously published mortality rates for patients with cancer. Inpatient mortality of patients with cancer treated with ICI was high in comparison with previously reported rates for hospitalized patients with cancer and was due to COVID-19 in almost half of the cases [9].
A cross sectional study in China suggests that lung cancer patients receiving ICI and experiencing irAEs may have a higher risk of developing COVID-19 pneumonia due to the Omicron variant. The study reports 20 patients on ICI therapy with Covid-19 pneumonia out of 72 patients with IRaes. Therefore, close monitoring of these patients during COVID-19 is necessary to mitigate this risk [20].
Further studies might state the role of other antifibrosant drugs in PF but at the moment a not delayed steroid treatment seems effective. PD-1 inhibitors immunotherapy and others ICI revolutionized lung and other cancers’ treatment. ILD is among adverse effects. We believe that Covid-19 might trigger or quicken PF arousal but the patient might have a predisposition to PF for ongoing Pembrolizumab therapy. Also cancer might predispose him to more severe Covid-19 manifestations. More extensive studies are needed to promptly manage patients and start valid treatment.
We found in literature few papers (4) about COVID-19 pneumonia overlapping ICI pneumonitis, so
we can say that ours is a rare case and it may be worth publishing it.
We added a paragraph and improved discussion in the text (lines 49-56, lines 144-145).
Line 50: Reference 6 can be deleted.
thank you, we delete the reference
Line 54: How was the COVID-19 diagnosis made? What test was used?
Thank you for the question, SARS-CoV2 RNA has been researched in all respiratory samples with qualitative real-time reverse-transcriptase polymerase-chainreaction (RT-PCR) assay. We added the information at line 78
Line 60: Line 60: What dose and how often was pembrolizumab administered?
Thank you for the question, dose of pembrolizumab was 200 mg/3 weeks
Line 60: What was the temperature value?
Thank you for the question, the temperature was 38.1 degrees. We added the information at line 72
Lines 65-69: The authors should provide more details about the tests performed on the patient. Other variables of interest should be described. Were bacteriological/mycological cultures performed on the patient? Were these studies negative?
For a better understanding of the article, the authors could make a diagram (figure) that addresses the time of study of the patient as well as the most relevant details.
Thank you for the question. Qualitative real-time reverse-transcriptase polymerase chain reaction (RT-PCR) tests for influenza A and B viruses, as well as respiratory syncytial virus, were negative. Blood tests for HIV 1 and 2 were also negative. Urinary antigens for Legionella and Pneumococcus returned negative results. Due to worsening clinical conditions, only microbiological data from sputum were available. Sputum examination was negative for common germs and mycobacteria.
|
|
|
|||||||
|
|||||||
Figure 1: Timeline shows the chronology of the imaging scans, COVID-19 swab results and therapies from diagnosis of cancer to COVID-19
We added the informations at line 77-81 and figure 1
Line 106: Line 106: Authors should add the following citation : Moffat GT, Hanna L, Hopman W, Fung AS, Gaudreau PO. An assessment of extended pembrolizumab dosing in advanced non-small-cell lung cancer in the COVID-19 pandemic. Immunotherapy. 2023 Aug;15(12):921-932. doi: 10.2217/imt-2022-0257. and discuss this idea in the article.
Thank you for suggestion, we added the citation in the bibliography
Line 109: Recently, Tanvetyanon T et al. described that the COVID-19 pandemic was associated with an increase in survival among patients with lower PD-L1 expression who were treated with pembrolizumab monotherapy. The authors should discuss this idea in their article.
In a recent study, Tanvetyanon Tand colleagues, describe the analisis of data derived from diverse oncology practices across the United States. They found that the outcome of patients who underwent frontline pembrolizumab-based treatment was impacted by the pandemic in some subgroup populations. Among those treated with pembrolizumab monotherapy, survival was better in the pandemic cohort than in the pre-pandemic cohort, when the PD-L1 expression level was <50%. However, when the PD-L1 level was ≥50%, the survival was not better in the pandemic cohort. This effect modification by PD-L1 expression level was not observed among those treated with pembrolizumab plus chemotherapy. The y suggest that it is plausible that there is a beneficial effect of SARS-CoV-2 infection in this specific population. Previous literature has suggested a possible synergistic effect of vaccination or viral infection with checkpoint inhibitor immunotherapy. It is hypothesized that vaccination can enhance the infiltration of central memory T cells into the tissues, leading to an enhanced anti-cancer immune response. The immune stimulatory effect due to SARS-CoV-2 infection may have been most beneficial among patients with tumours expressing lower PD-L1 undergoing pembrolizumab monotherapy.
However, the study is limited by its non-randomized design, and the data were obtained during routine clinical care, not a controlled research environment, and thus subjected to delayed or missing data.
The case described in our paper could support the researchers' hypothesis, as the clinical outcome of the patient was favourable. However, further prospective studies are necessary to confirm the hypothesis.
We added the suggestion in discussion at lines 158-163.
Line 149: The article's conclusions are redundant and very long. There are elements that could be incorporated into the discussion of the work. The conclusions should be rewritten.
Thank you, we modified discussion and conclusion.
Line 232: Reference 10 must be updated.
Thank you, we added the following citation :
Hu H, Wei S, Huang J, Sharma L, Chang D. Beneficial effects of early low-dose methylprednisolone with long-term treatment in ARDS. Eur J Intern Med. 2024 Nov;129:137-139. doi: 10.1016/j.ejim.2024.06.009. Epub 2024 Jun 15. PMID: 38879352.

Reviewer 3 Report
Comments and Suggestions for Authors
This paper presents a detailed case study on a patient treated with pembrolizumab for non-small cell lung cancer who later developed COVID-19 pneumonia and pulmonary fibrosis. This is an interesting case as it examines an important intersection—cancer immunotherapy and COVID-19 complications—which has implications for both oncology and infectious disease management. I have the following comments and suggestions:
- Please correct the abbreviation list at the end of the manuscript so they appear in alphabetical order.
- The references do not follow the instructions to authors.
- There are at least three different font types in the manuscript.
- There is a discrepancy in COVID naming (COVID-19 vs Covid 19).
- Provide the full name of COVID-19 the first time it is mentioned.
- Line 59: Replace “image 1” with “figure 1.”
- Likewise, replace “image 2” (line 65) and “image 3” (line 81) with “figure 2” and “figure 3,” respectively.
- In light of the above comment, I suggest merging all the panels of all figures into one figure and providing, on top of each time point (A, B, and C), the date. In addition, in figure 2—which will be panel B in the revised manuscript—I suggest adding arrowheads to indicate the “ground glass” and “crazy paving” areas. Likewise, in figure 1 (panel A in the revised manuscript), an arrowhead should point to the tumor mass of 24×16 mm, etc. In the revised manuscript, the authors should refer to each time point as “as shown in Figure 1A” or “(Figure 1A),” etc.
- Line 83: Delete “2.1 Figures,” because a manuscript cannot have a section title labeled “Figures.”
- I suggest the authors follow the instructions to authors carefully.
- Despite the authors mention the need for additional research, they do not provide sufficient insights into long-term patient outcomes and potential guidelines.
- The overlap between pembrolizumab-related pneumonitis and COVID-19 pneumonia is acknowledged However, the authors do not dive deeply into specific differential diagnostic strategies.
Author Response
Rev 3: conclusions must be improved
This paper presents a detailed case study on a patient treated with pembrolizumab for non-small cell lung cancer who later developed COVID-19 pneumonia and pulmonary fibrosis. This is an interesting case as it examines an important intersection—cancer immunotherapy and COVID-19 complications—which has implications for both oncology and infectious disease management. I have the following comments and suggestions:
- Please correct the abbreviation list at the end of the manuscript so they appear in alphabetical order.
Thank you for the suggestion, we modified accordingly the paper
- The references do not follow the instructions to authors.
Thank you for the suggestion, we modified accordingly the paper
- There are at least three different font types in the manuscript.
Thank you for the suggestion, we modified accordingly the paper
- There is a discrepancy in COVID naming (COVID-19 vs Covid 19).
Thank you for the suggestion, we modified accordingly the paper
- Provide the full name of COVID-19 the first time it is mentioned.
Thank you for the suggestion, we modified accordingly the paper
- Line 59: Replace “image 1” with “figure 1.”
Thank you for the suggestion, we modified accordingly the paper
- Likewise, replace “image 2” (line 65) and “image 3” (line 81) with “figure 2” and “figure 3,” respectively.
Thank you for the suggestion, we modified accordingly the paper
- In light of the above comment, I suggest merging all the panels of all figures into one figure and providing, on top of each time point (A, B, and C), the date. In addition, in figure 2—which will be panel B in the revised manuscript—I suggest adding arrowheads to indicate the “ground glass” and “crazy paving” areas. Likewise, in figure 1 (panel A in the revised manuscript), an arrowhead should point to the tumor mass of 24×16 mm, etc. In the revised manuscript, the authors should refer to each time point as “as shown in Figure 1A” or “(Figure 1A),” etc.
Thank you for the suggestion, we modified accordingly the paper
- Line 83: Delete “2.1 Figures,” because a manuscript cannot have a section title labeled “Figures.”
Thank you for the suggestion, we modified accordingly the paper
- I suggest the authors follow the instructions to authors carefully.
Thank you for the suggestion
- Despite the authors mention the need for additional research, they do not provide sufficient insights into long-term patient outcomes and potential guidelines.
Thanks for your question,
the patient discontinued pembrolizumab after PF events (after 37 months of therapy). The patient is alive and in complete remission 18 months after stopping therapy.
Immune checkpoint inhibitors (ICIs), particularly PD-1 inhibitors, are known to cause pneumonitis, with a higher risk observed in patients receiving combination therapy compared to monotherapy. The severity of pneumonitis depends on tumor type and treatment regimen, with non-small cell lung carcinoma (NSCLC) patients being at particularly high risk. Smoking, underlying lung conditions, and genetic mutations like EGFR also contribute to this risk. The differential diagnosis between ICI-related pneumonitis and COVID-19 pneumonia is complex due to overlapping symptoms and radiological features. False-negative upper airway tests further complicate this, highlighting the need for early bronchoscopic evaluation and potential presumptive treatment in cancer patients.
The pandemic has exacerbated the challenges faced by cancer patients undergoing immunotherapy, with data showing poor outcomes for those treated within 1-3 months prior to contracting COVID-19. Despite conflicting evidence on immunosuppression with ICIs, they should not be avoided due to hypothetical risks of severe COVID-19. COVID-19 vaccination remains a priority for cancer patients, with those receiving immunotherapy expected to develop protective immunity despite potential risks of exaggerated immune responses. Oncologists must make informed decisions, balancing the benefits of ICIs against the risks posed by COVID-19, and ensure that both providers and patients understand these risks. Patients should be vaccinated, and strict mitigation strategies should be maintained in cancer centers to minimize nosocomial transmission.
Prompt NPS execution for fever or inflammatory symptoms and serum inflammation indices can aid differential diagnosis. Guidelines are needed for managing cases of COVID-19 pneumonia and pembrolizumab-induced pneumonitis, including immediate systemic steroid therapy, which benefited our patient.
We added the suggestion in the discussion in lines…..and in lines
- The overlap between pembrolizumab-related pneumonitis and COVID-19 pneumonia is acknowledged However, the authors do not dive deeply into specific differential diagnostic strategies.
Thank you for the suggestion.
Chest CT findings in COVID-19 patients commonly include peripheral, bilateral or multifocal GGOs with or without consolidation near pleural surfaces, fissures, or visible interlobular lines, creating a ‘crazy paving’ pattern. Later stages may show reverse halo and other organising pneumonia signs[Chung M, Bernheim A, Mei X, Zhang N, Huang M, Zeng X, et al. CT imaging features of 2019 novel coronavirus (2019‑nCoV). Radiology 2020;295:202‑7].
CT findings of ICI-related pneumonitis vary, often showing patchy GGOs and consolidation, symmetrical distribution, predominantly in lower lobes and peripheral areas. Patterns can include non-specific interstitial pneumonia, organising pneumonia, diffuse alveolar damage, hypersensitivity pneumonitis, or simple pulmonary eosinophilia, overlapping with COVID-19 pneumonia
Criteria for diagnosing drug-related pneumonitis include: (a) newly detected parenchymal opacities on CT or chest radiographs, usually bilateral and non-segmental; (b) temporal association with a systemic therapeutic agent; and (c) exclusion of other causes [Johkoh T, Lee KS, Nishino M, Travis WD, Ryu JH, Lee HY, et al. Chest CT diagnosis and clinical management of drug‑related pneumonitis in patients receiving molecular targeting agents and immune checkpoint inhibitors: A position paper from the Fleischner Society. Radiology 2021;298:550‑66].
As COVID-19 becomes endemic, differentiating it from other causes will be increasingly challenging. Understanding GGOs differentials is crucial for guiding treatment and improving patient outcomes.
We observed the patient developed fibrotic-like patterns with consolidation in lower lobes and worsening crazy paving areas.
Due to the overlap in radiological appearances, clinical suspicion and testing for SARS-CoV-2 and other respiratory viruses should be performed for lung deterioration in IC-treated patients .
We added the suggestion in the discussion in lines 168-174 and 178-183.

Round 2
Reviewer 2 Report
Comments and Suggestions for Authors
The authors have met the reviewer's requirements. I accept the work.